# A qualitative study of patients' feedback about Outpatient Parenteral Antimicrobial Therapy (OPAT) services in Northern England: implications for service improvement

Maureen Twiddy,[1] Carolyn J Czoski Murray,[2] Samantha J Mason,[3] David Meads,[2] Judy M Wright,[2] Elizabeth D Mitchell,[1] Jane Minton,[4] on behalf of the CIVAS study team

[1]Hull York Medical School, University of Hull, Hull, UK
[2]Leeds Institute of Health Sciences, University of Leeds, Leeds, UK
[3]Leeds Institute of Cancer and Pathology, University of Leeds, Leeds, UK
[4]Leeds Teaching Hospitals NHS Trust, Leeds, UK

**Correspondence to**
Dr Maureen Twiddy;
maureen.twiddy@hyms.ac.uk

## ABSTRACT

**Objective** Outpatient parenteral antimicrobial therapy (OPAT) provides opportunities for improved cost savings, but in the UK, implementation is patchy and a variety of service models are in use. The slow uptake in the UK and Europe is due to a number of clinical, financial and logistical issues, including concern about patient safety. The measurement of patient experience data is commonly used to inform commissioning decisions, but these focus on functional aspects of services and fail to examine the relational aspects of care. This qualitative study examines patients' experiences of OPAT.

**Design** In-depth, semistructured interviews.

**Setting** Purposive sample of OPAT patients recruited from four acute National Health Service (NHS) Trusts in Northern England. These NHS Trusts between them represented both well-established and recently set-up services running nurse at home, hospital outpatient and/or self-administration models.

**Participants** We undertook 28 semistructured interviews and one focus group (n=4).

**Results** Despite good patient outcomes, experiences were coloured by patients' personal situation and material circumstances. Many found looking after themselves at home more difficult than they expected, while others continued to work despite their infection. Expensive car parking, late running services and the inconvenience of waiting in for the nurse to arrive frustrated patients, while efficient services, staffed by nurses with the specialist skills needed to manage intravenous treatment had the opposite effect. Many patients felt a local, general practitioner or community health centre based service would resolve many of the practical difficulties that made OPAT inconvenient. Patients could find OPAT anxiety provoking but this could be ameliorated by staff taking the time to reassure patients and provide tailored information.

**Conclusion** Services configurations must accommodate the diversity of the local population. Poor communication can leave patients lacking the confidence needed to be a competent collaborator in their own care and affect their perceptions of the service.

### Strengths and limitations of this study

► We recruited from four diverse sites, which enabled us to contrast the views of those who experienced different models of care.
► A relatively large qualitative sample (n=32) patients were interviewed and our broad sampling strategy meant we obtained views from participants from a diverse range of socioeconomic backgrounds.
► The views of the very elderly and those from ethnic minority groups are not well represented.
► Data collection continued after data saturation was reached.

## INTRODUCTION

Outpatient parenteral antimicrobial therapy (OPAT) allows patients to be given intravenous antibiotics while living in the community, rather than as a hospital inpatient. It can be used for patients with a range of infections, but most commonly used for complicated soft tissue infections, bone and joint infections, endocarditis and bacteraemia[1] Although standard practice in many countries, uptake of OPAT in the UK has been slower,[1 2] and hampered by a range of clinical, financial and logistical issues, not least the lack of a national commissioning strategy to support its expansion.[2 3]

Three service models can be used to deliver OPAT: outpatient/ambulatory care centres; a nurse visiting the patient at home or the patient/carer trained to self-administer. With a national focus on efficiency savings and improving patient experience, OPAT is becoming more popular.[1] It is therefore important to understand patient experiences of different OPAT services to inform commissioning decisions.

In recent years UK health policy has used measures of patient experience to identify strengths and weaknesses of service provision to inform commissioning, determine resource allocation and drive up quality.[4] Across a range of clinical conditions, studies have found positive associations between patient experience (defined by National Institute for Health and Care Excellence and the Institute of Medicine as the relational and functional aspects of care)[5][6] and a range of quality indicators, including patient safety and clinical effectiveness[4][7] However, the type of data collected by these surveys do not help us understand what matters to patients,[8] and tells us little about what good care looks like. This study explores patients' experiences of OPAT services to identify issues that affect patient experience and satisfaction.

### What is already known?

Evidence regarding patient experiences of OPAT has been largely collected using patient satisfaction surveys, and there is little qualitative research to illuminate patients' experiences of these services. The results of our systematic review of OPAT services[9] found patients identified a range of benefits such as the comfort of the home environment and increased freedom and autonomy, but not all patients view OPAT positively, with safety concerns reported.[10] There is also evidence to suggest that the information needs of OPAT patients may not always be addressed and some may find OPAT anxiety provoking.[10] This suggests that both functional and relational aspects of patient experience are important, but the dearth of evidence makes it difficult to draw conclusions on what good and poor care look like.

This study sought to understand patient experiences of OPAT to identify what was important to them and is part of a larger programme of work. The interviews were used to develop a discrete choice experiment to examine patient preferences for services.[3] This paper constitutes a reanalysis of these data to examine patient experiences more broadly.

## METHODS
### Design

Semi structured interviews and focus groups.

### Study settings

Four hospitals in Northern England were purposively selected as they offered the following three care pathways: hospital outpatient attendance, nurse at home and self-administration (table 1).

### Participants

A purposive sampling strategy identified two groups: patients requiring short-term intravenous antimicrobials (<7 days: n=15) and patients with deep-seated infections requiring longer-term intravenous antimicrobials >14 days; n=25). The sample size assumed those on longer term antimicrobials represented a wider range of infections.

**Table 1** Local site characteristics

| Site | Population size | Services provided |
|---|---|---|
| Teaching hospital | 500 000 | Well-established outpatient attendance, visiting general nurse and self-administration. |
| Teaching hospital | 800 000 | Well-established outpatient attendance and self-administration. New visiting specialised nurse service. |
| District general hospital | 330 000 | Well-established visiting specialised nurse, outpatient attendance and self-administration. |
| District general hospital | 385 000 | Newly established service offering visiting general nurse and outpatient attendance. |

A sampling frame was developed to capture variation in age, gender and socioeconomic status. Initially, focus groups were planned but these proved difficult to recruit to so interviews were offered.

### Procedure

Interviews took place at the patient's home or the university and were conducted by MT, CJCM or SJM (who have backgrounds in psychology, sociology and nursing). Patients were consented by nursing staff so the only contact researchers had with the participant was during the interview. The focus group was facilitated by MT and CJCM and took place on National Health Service (NHS) premises. Home interviews adhered to the University lone working policy to ensure staff safety. Written informed consent was obtained for all participants, and discussions audio-recorded, with permission. Participants were informed of their right to withdraw at any time; no participants withdrew. One participant refused to be recorded (notes were taken). Interviews lasted between 30 and 75 min; the focus group lasted 95 min.

### Topic guide

The topic guide covered three questions, with probes used to explore issues in more detail. The topic guide was initially piloted on three patients and no changes made:

▶ What has been your experience of OPAT? What were the good and bad points in the care/service you received?
▶ What are the most important aspects of intravenous antibiotic services for you?
▶ If you were designing a service to provide community antibiotic intravenous services what would it look like?

### Data analysis

The interviews were originally conducted as part of a mixed methods study to identify attributes of care which could be used to develop a discrete choice experiment.[3] This paper provides a reanalysis of that those data to

understand participant experiences and in doing so takes a subtle realist approach which accepts the social world exists independently of our understanding of it, but that it is only accessible via participants experiences and interpretations.[11]

Audio recordings were transcribed verbatim, anonymised and managed using NVivo10 software.[12] Transcripts were checked to ensure patient confidentiality was maintained and material removed from the transcript which could possibly identify individuals (eg, name of their doctor, family members). Coding was inductive, identifying issues of importance to patients. Data were later explored using the conceptual framework developed by Entwistle and colleagues.[13] Two researchers (MT, SJM) independently read and coded the first three interviews. This became the initial coding frame. Codes were sorted into categories based on how they relate to one another, and themes formed. The research team agreed the coding index which was then applied to the remaining transcripts by one researcher (SJM). Data saturation was reached as no new ideas were identified from the last five interviews.[14] Interview transcripts were requested by three participants; no requests for changes were received. The following notation is used in the quotes […]=text omitted. Quotes indicate participant gender, age group, course of antimicrobial (short term/long term) and model of care experienced (nurse at home, self-administration and hospital outpatient clinic).

## RESULTS

A total of 41 patients consented. Nine subsequently declined participation preinterview due to illness or could not be contacted. One focus group (four participants) and 28 interviews took place. One interview was not used as the participant did not recall having OPAT. The focus group participants came from one hospital and all received a nurse at home model so although the issues they identified around nurse at home care reflect the experiences of patients at other centres they did not contribute to our understanding of the other models of care. As a result the findings of the focus group and interviews were analysed with the interview data. Demographic details are in table 2.

Two key themes were identified which map to functional and relational aspects of care.

### Functional aspects of care

This theme relates to the functional aspects of care which are described by four subthemes: being home, but not well; convenience and flexibility; location of care; is it safe?

### Being home, but not well

For most patients, OPAT was an opportunity to be discharged from hospital earlier than would otherwise be the case. These participants believed that recovering at home would be better than being in hospital and

**Table 2** Participant demographics

|  | n = 32 |
| --- | --- |
| Age | Mean = 53 years (range 21–80) |
| Gender, male | 16 |
| Marital status | |
| Married | 16 |
| Single | 7 |
| Divorced/separated/widowed | 3 |
| Cohabiting/civil partnership | 6 |
| Ethnicity | |
| White British | 29 |
| White European | 2 |
| Other (not stated) | 1 |
| Education | |
| University/professional qualification | 14 |
| College | 9 |
| Secondary | 7 |
| Did not complete formal education | 2 |
| Employment | |
| Full time (>30 hours/week) | 12 |
| Part time (<30 hours/week) | 4 |
| Unable to work due to ill health | 5 |
| Retired | 10 |
| Carer | 1 |
| Infection type | |
| Short term/long term | 20/12 |
| Service received | |
| Hospital outpatient | 14 |
| Nurse at home | 13 |
| Self-administration | 5 |

welcomed the opportunity to try OPAT. However, few realised how difficult it would be to look after themselves at home and some felt staff should have been more alert to their personal situation and circumstances.

> I never realised how tiring it would be though […] I never realised that just making a cuppa could be so tiring (female, age 60–70, long term, nurse at home)

> I've just begun starting to pick up tasks again, I'm not quite there yet where I'm a fully functioning mum (female, age 40–50, long term, hospital outpatient)

For others, OPAT was an opportunity to avoid hospital admission, and although some people continued to work, for others, the infection limited their activities.

> All I went and did, was, go in the car, go to the hospital and come home, and I didn't go anywhere else, […] the first three days I felt really, really, ill, so I

didn't want to go anywhere or do anything (female, age 50–60, short term, hospital outpatient)

## Convenience and flexibility

Although some hospital outpatient OPAT services were managed via an appointment system, one NHS Trust ran their OPAT service from the medical admissions ward, leading to significant delays which proved particularly difficult for patients who were trying to balance going to work and treatment.

[I] thought 'I'll be back in work by sort of quarter to eleven', by three o'clock I still hadn't been seen, […] I didn't like that whatsoever (male, 40–50, short term, hospital outpatient)

Despite having an infection serious enough to require intravenous treatment many working age people did not take sick leave. Some felt well, but others found it difficult because they were not viewed by their managers as 'ill enough'.

I can't walk, booked a week and a half off work, […] then my boss rang me and was like 'I need you to work' (female, age 30–40, short term, hospital outpatient)

When appointment systems worked well most found hospital attendance convenient and appreciated that treatment could be fitted around their personal circumstances.

[coming to hospital] it's better for my employers (male, age 40–50, short term, hospital outpatient)

For patients who self-administered, multiple treatments each day can leave them with little time to fit anything else into their day. Although some coped by taking their intravenous kit with them and infusing 'on the go', others found the perceived benefit of being at home was eroded, as planning the next treatment was always at the back of their mind.

There's no point really going out much or doing much cos you haven't got much time when you aren't having to think about getting everything sorted (male, age 30–40, long term self-administer)

## Location of care

Where care is delivered was important to patients. Travelling to hospital could be challenging for those who relied on public transport, when apparently 'short' distances could result in two or three bus changes. Even travelling by car, patients found it difficult to park and fees quickly mounted. Some patients suggested that dedicated short-term parking bays, similar to those used by dialysis patients would help alleviate these issues. Cognisant of the cost of a nurse visiting them at home some patients suggested that OPAT services could be located in general practice health centres. Others would have liked to visit their general practitioner practice because they found the nurse at home model too restrictive.

waiting in for a district nurse wasn't something that I liked, because of the inconvenience of being tied to your home waiting for them (female, age 50–60, short term, hospital outpatient)

[older people] feel a little bit more scared of the hospitals because some of them are single or widowed so they don't always have somebody to go with them […] a local clinic would be much less stressful for them (male, age 50–60, short term, hospital outpatient)

It was important that the OPAT model offered to patients met their needs. For those with multimorbidities, attending the hospital daily or three times a week for treatment was viewed more negatively than being an inpatient, making a nurse at home model necessary. These patients also often had multiple agencies involved in their care and so it was important to ensure they could cope in the community as they were often weakened by the effect of other conditions.

I don't think that [clinic] would have worked, because […], I was still extremely weak, […] To physically have to make a journey each day, un-necessarily in my eyes, because if I'd have stayed in hospital I wouldn't have had to make the journey, […] that done would have been exhausting (female, age 40–50, long term, specialist nurse)

## Is it safe?

Safety combined both functional and relational aspects of care. Concerns about infection risk are acknowledged and described by patients. Patients expressed confidence in the staff working in the service to minimise risks, and talked of the professionalism they had observed.

nurses […] were very knowledgeable about what I was experiencing and this reassured me about coping at home (female, age 40–50, long term, specialist nurse)

For some, the hospital was viewed as a safe place to receive their treatment because doctors were in attendance at the clinic, and for these patients, this embodied, *'a safe service'*. Although they were treated by a nurse, knowing a doctor was in attendance and able to monitor their care, was an attractive safety net, due to their perceived increased expertise.

… so personally for me I felt like being treated at the hospital was probably the best option because there'd have been people around who could have come and had a look at me if they'd needed to (female, age 50–60, short term, hospital outpatient)

The nurse at home model was perceived to be a safe service because it minimised the risk of contracting infections such as *Clostridium difficile* (C Diff) which they associated with hospital attendance. For a few patients the perceived benefits of hospital attendance did not entirely dispel these worries, and over one-third of patients made

some reference to the risks associated with methicillin-resistant *Staphylococcus aureus* (MRSA).

> I just thought I would end up getting C Diff or MRSA in my leg. I don't want to be laid up and I don't feel ill. (female, age 40–50, short-term, hospital outpatient)

For those patients self-administering their intravenous treatment at home, the concepts of safety and risk were more complex. All were at significant risk of contracting infections due to underlying health issues, and knew being at home reduced this risk. For them, maintaining aseptic technique, correctly storing their medication and administering their drugs were second nature, but all were aware of the consequences of any lapse of judgement and valued the reminders given by nursing staff.

> They give you a booklet every single time and go through it every time; they go through obviously your flushes, even though we've been doing it for years, cos if I get it wrong I am back in here (hospital) (female, age 20–30, long term, self-administration)

### Relational aspects of care

This theme describes the relational aspects of care, such as emotional support, treating people as individuals, good communication and information which were key to good quality care. Participants gave examples of where nursing staff had reduced patients' OPAT related anxiety and distress, and explained that they did this with sensitivity and professionalism, ensuring that the patient's dignity was maintained.

> I felt, I felt quite sorry for them cause I was just having such a panic and just like, you must have to deal with crazy people all the time and they were really nice […], they didn't make a big thing of it but got me somewhere quiet (female, age 20–30, short term, hospital outpatient)

All patients recalled receiving good quality written information but this was often generic and did not answer their questions; for example, how to shower with a cannula in place, or how to get additional support at home. Some older patients had concerns about being cared for out of hospital, and described how having the nurse to talk to provided the confidence needed to self-manage.

> I'd got that attention completely for that time […]. you've just got their attention no matter what, you get to know them. I found them easy then to open up to, to ask questions (female, age 60–70, long term, specialist nurse)

Although there were many examples of good care, the presence of a cannula or port to facilitate drug administration was distressing for many who had no previous experience of intravenous administration, and patients felt that staff did not appear to acknowledge the impact this had on them, in particular the fear it engendered going about everyday activities.

> They gave me, you know the cannula, they were like 'we put this in and we leave it in your arm' which made me like die a little bit inside, then, the fear of it being knocked at home, that killed me (female, age 30–40, short term, hospital outpatient)

The visibility of the cannula was particularly troublesome when travelling by public transport as there was a perceived risk of injury and a fear of being judged by its presence.

> I thought y'know err what's people gonna think about this? [I was] concerned about how it would be perceived you know, wandering [about with cannula in] (female, age 40–50, short term, hospital outpatient)

In contrast, self-administration patients had formal training about intravenous management and access to a nurse by phone to provide ongoing support which they viewed as essential and enabled them to be fully involved in decisions about their care.

A perceived breakdown in communication between OPAT staff could erode patient confidence, and fuel anxiety about not being in hospital. When, on one occasion a nurse arrived not knowing they were to give an intravenous treatment, the patient questioned the competence of the team. Similarly, examples were given of staff coming to remove a cannula that had already been removed, or to give intravenous antibiotics to patients who had been switched to oral medication and these were provided as examples of poor care.

> She had no clue who I was really and arrived not knowing that she was supposed to bring the drugs with her, it did make me wonder about them (female, age 40–50, long term, district nurse)

A key transition in terms of patient care was at the end of intravenous treatment. Patients with long-term infections were reviewed regularly, and seen at the end of treatment, and all were satisfied with their follow-up.

> I've got follow up in a month which is nice so they're keeping an eye on me, I wouldn't like it if I hadn't been (female, age 40–50, long term, hospital outpatient)

In contrast, short-term intravenous patients were not seen in clinic again and some were given no advice about what to do if symptoms returned. Although a discharge letter was sent to the patients' general medical practitioner, few patients were aware of this and even fewer knew whose responsibility it was to organise a follow-up appointment if needed. This lack of continuity of care was most evident with patients who had been cared for by a nurse at home as they had generally not seen a doctor after the initial diagnosis, and these patients commonly described feeling left in the dark about their future care.

> I was left in the dark as to know what was after the IV, nothing at all. I'd rather if they said ok, make an

appointment to see your doctor (male, age 50–60, short-term, specialist nurse)

For these short-term intravenous patients the end of treatment was a key point where things could, and did, go wrong, and the lack of clarity about what should happen next caused uncertainty as patients were unsure who to contact.

The doctor said four weeks when I saw her, but I'm more than four weeks on from seeing the doctor and it's still not entirely right so I don't know, no-one told me anything (male, age 40–50 short term, hospital outpatient)

## DISCUSSION

Patients identified a range of healthcare experiences as important to the quality of care received. Important considerations were: being cared for in a way that fits their personal circumstances (location and type of OPAT), the type of staff involved and staff able to deliver good quality care. Where patients were cared for and by whom was important. For some, this meant doctors being visibly involved in service delivery; for others a nurse led service was appropriate. All participants recognised that nurses' ability to recognise and respond appropriately to changes in the patient's health contributed to a positive healthcare experience.

Satisfaction with OPAT services was high, a finding which is well reported in the literature.[9] However, there was also evidence that services were not always well aligned to the personal and material resources of the patient. The contextual factors that affected how well patients cope included: what support families had at home, personal circumstances (eg, self-employed), material resources, such as car access for daily attendance at hospital and the provision of information tailored to their situation. Other studies have found families may not have the personal resources to care for a family member at home,[15 16] and our findings support this conclusion.

It is recognised that patients often find being cared for out of hospital worrying, and providing access to advice can boost confidence.[16 17] However, the information needs of patients are often not met.[10] The present study supports these findings, but also suggests that even when patient outcomes are good, as was the case in this study, interpersonal relationships are important.

Most patients were provided with good written information but tailored information was absent, and oral communication between patients and staff was more variable. Positive relationships developed when staff found time to talk to patients about their treatment and understand them as people, rather than cases. These encounters could help patients develop the confidence needed to take a more active role in their own care. Poor communication left some without the knowledge and confidence needed to be actively involved in their own

care, and affected their perceptions of the service. These findings resonate with the conclusions of a recent review by Entwistle and colleagues[13] looking at the aspects of healthcare delivery that are most important to patients. Entwistle's study suggests that both the structure of healthcare and the social dynamics are important to the patient experience. Our findings lend support this conclusion.

The perceived risk of contracting a hospital acquired infection was at the forefront of the minds of many patients. With the media labelling MRSA a superbug, it is not surprising that patient perceptions of the risk of contracting an infection have not yet caught up with the reality of reducing cases of MRSA. Earlier studies have found there to be high levels of awareness of MRSA, with one study in 2006 finding 94% of patients were aware of MRSA, with 68% finding information about MRSA from the media.[18] Another qualitative study found the majority of patients had little confidence in the NHS in relation to healthcare-related infections.[19] In the present study, this discourse was still prevalent and suggests more needs to be done to educate the public about the actual risk of MRSA, and how to minimise these, especially in the light of increasing drug resistance.

The findings of this study were used to develop a discrete choice experiment (DCE) to seek to understand patient preferences for OPAT services. The DCE was distributed to 202 people who had previous experience of OPAT and found that looking at the whole sample, patients were more likely to choose a nurse at home model over a hospital or self-administration model; there was a preference for timed appointments, and for treatment delivered by a specialist, rather than generalist nurse, and communication with someone they know. However, there was significant heterogeneity across patient types, although with an overall preference for the nurse at home model.[3] These findings align with our qualitative findings and argue for flexible service as a one-size does not fit all.

### Strengths and limitations

Our data support and develops the previously limited qualitative research evaluating OPAT services. OPAT can allow patients to receive care in the community but patient satisfaction can be reduced if not configured to the local population. A strength of this study is that we recruited from four diverse sites,[3 20] and a broad sampling strategy was used to obtain views from participants from a diverse range of socioeconomic backgrounds. However, we struggled to recruit the very elderly and those from ethnic minority groups. We planned to undertake focus groups but recruitment was poor, so we switched to interviews which resulted in the data collection continuing after data saturation was reached, and so no new findings were revealed in the final five interviews.

### CONCLUSION

Nationally and internationally, healthcare organisations have highlighted the importance of patients' experiences

of the services they receive, and indeed the NHS Operating Framework for England (2011) describes each patient's experience as 'the final arbiter of everything the NHS does'.[21] In the current drive to have patients cared for in the community it is important to ensure that services are designed in a way which meet the needs of the local community to improve patient's experiences of healthcare delivery.[22] This study shows poor communication can leave patients lacking the confidence needed to be a competent collaborator in their own care, and affect their perceptions of the service, even when they have positive health outcomes.

It is therefore important to understand what aspects of service provision are most important to the patient, in order to improve services.

**Acknowledgements** The authors thank the rest of the CIVAS research team, our patient and public involvement group and all the sites for their support making this research project a success.

**Collaborators** Stephane Hess, David (Theo) Raynor, Rachel Vincent, Philip Stanley, Kate McLintock, Angela Gregson.

**Contributors** MT, CJCM, DM and JM designed the study. MT, CJCM and SJM undertook the data collection and interpreted the data. EDM and JW undertook the literature searches supporting the study. All authors participated in writing the manuscript, and read and approved the final version.

**Funding** This work is supported by the National Institute for Health Research HS & DR Programme (grant no 11/2003/60).

**Competing interests** None declared.

**Patient consent** Obtained.

**Ethics approval** Ethical approval was sought and obtained from NRES Committee South West—Frenchay (13/SW/0060).

**Provenance and peer review** Not commissioned; externally peer reviewed.

**Data sharing statement** No additional data are available from this study.

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
