## [Reviewer comments · BMJ Open]

ARTICLE DETAILS

TITLE (PROVISIONAL)	Listening to patients' feedback about Outpatient Parenteral Antimicrobial Therapy (OPAT) services – implications for service improvement
AUTHORS	Twiddy, Maureen; Czoski Murray, Carolyn; Mason, Samantha; Meads, David; Wright, Judy; Mitchell, Elizabeth; Minton, Jane

VERSION 1 – REVIEW

REVIEWER	Dr Antonella Tonna Lecturer in Clinical Pharmacy Robert Gordon University Aberdeen AB10 7GJ United Kingdom
REVIEW RETURNED	18-Sep-2017

GENERAL COMMENTS	This is a very interesting topic and very encouraging that patient's aspect and perspective is being researched in depth using qualitative methodology. Agree with authors that this allows for in-depth findings. May have been more clear that potentially this is an exploratory phase aimed at information a second phase - the survey. Authors' list - please review - unclear whether full list of authors has been provided here. Please find comments below. Abstract: Results: may need to have some more details on actual results - patient opinions not included. The results in the abstract section do not seem to reflect the results in the full text. Please review. Introduction: Para 2 and 3: better placed at start and leading up to aim of this study. Please double check accuracy of referencing: e.g. Ref 11 refers to systematic review but unsure if ref at end list is systematic review. Some references are a little dated. Methods: Some more details required here - e.g. who conducted the interviews? Some more details required on ethics - patient confidentiality, safety at home. Results: More details required describing source of quotes e.g. <65 and >65 may be too broad. This will help contextualise quotes further.
---

	Would also help reader determine the broadness and range of participants included. Timeliness of service: Lines 16-23; need quote to support the report re complaints. Some quotes very long - may be beneficial to cut down. Clarification on which are themes/subthemes particularly in the "delivery of care" section. Also does this relate to functional aspects which are not referred to in results section of text. Care with bacterial nomenclature = P13. Line 47 - medical equipment breakdown - not relating to OPAT. Unclear whether this influenced patient choice. Relational aspects: There is plenty of overlap between 2 themes. Authors may consider whether more informative to combine the two together. Also aspects such as "dignity" referred to at introduction which do not seem to be followed through. Any difference between themes from interviews and focus groups? Discussion: Needs expansion. Most is a review summarising key findings. Only ref 26 referred to in appropriate depth. Avoid use of quantitative descriptors when analysing qualitative data - e.g. "over half of patients"; "1/3 of patients" Strengths: claims that contrasts views of different experiences - this may be clearer in results section. Also needs more discussion on how study was used or may be used to inform practice and policy development.
--	--

REVIEWER	Trine Bernholdt Rasmussen Dept of Cardiology, Herlev and Gentofte Universityhospital, Copenhagen, Denmark
REVIEW RETURNED	03-Oct-2017

GENERAL COMMENTS	OPAT services have the potential to reduce health care cost and improve patients' well-being during treatment, but little is indeed known about the patients' perspective; which model of service they prefer, what are potential challenges and possible improvements. The work presented in this manuscript is indeed important and relevant. I find the manuscript to be very well written, concise and clear. I have only one major and a few minor concerns/suggestion: Major: To me the structure in the presentation of the findings is very confusing. What are major themes and what are sub-themes? In the abstract it seems functional and relational aspects of care are the two major themes with three and two sub-themes respectively? In the result section however I cannot find this pattern. It says (under a heading called Delivery of care...is this I wonder Functional aspects of care?): "Three main issues around delivery of OPAT were identified..." which are?? Are main issues the same as major themes? Then "These are described by four subthemes: being home, but not well; timeliness of care; location of care; is it safe?"...but there were only three sub-themes in the abstract. On page 14 the theme Relational aspects of care with two sub-themes is presented (in concordance with the abstract)... is that then one of the three main issues around delivery of OPAT...and where is the third?. It seems also that different categories/wording is used. Is Timely management of symptoms (in abstract) for instance, the same as Timeliness of care (in manuscript)? Authors should work through this critically, so it is clear to the reader what the identified themes and sub-themes are and how they inter-relate.
--

	I would suggest using headings and subheadings in the result section to make it clear. Also of course consistency between the categories and wording in the abstract and the manuscript. Minor: P. 5 line 12 on the association. Patient experience can be good or bad, is it not rather patient preference? The references – I believe - are ‘misplaced’. It looks to me as though a reference was omitted and then the rank order wasn’t changed. Authors should review references thoroughly and make sure they are correct, complete and updated. I would also like a few explanatory sentences about the methodological framework chosen and why, before the description of the actual thematic analysis performed.
--	--

VERSION 1 – AUTHOR RESPONSE

Reviewer #1

This is a very interesting topic and very encouraging that patient's aspect and perspective is being researched in depth using qualitative methodology.

Agree with authors that this allows for in-depth findings. May have been more clear that potentially this is an exploratory phase aimed at information a second phase - the survey.

Response: We do have a sentence at the end of the introduction which states this is part of larger piece of work. We have expanded on this to explain this work was undertaken to inform the development of a large scale survey and referenced our NIHR report which includes the DCE findings.

This study sought to understand patient experiences of OPAT to identify what was important to them and is part of a larger programme of work. The interviews were used to develop a discrete choice experiment to examine patient preferences for services.³ This paper constitutes a reanalysis of these data to examine patient experiences more broadly. (lines 117-120)

Authors' list - please review - unclear whether full list of authors has been provided here

Response: The authors listed comprise those who fulfil the obligations and requirements for authorship on the paper and do not constitute the full research team. We have made it clearer in the authorship that we are publishing this paper on behalf of a larger research team. All team members are acknowledged in the paper. (lines 1-9 and 478-481)

Abstract: Results: may need to have some more details on actual results - patient opinions not included. The results in the abstract section do not seem to reflect the results in the full text. Please review.

Response: We have taken on board the reviewers comments about the links between the abstract results and main body text and completely rewritten the abstract results section. See lines 48-62

Results: Despite good patient outcomes, experiences were coloured by patients' personal situation and material circumstances. Many found looking after themselves at home more difficult than they expected, whilst others continued to work despite their infection.

Expensive car parking, late running services and the inconvenience of waiting in for the nurse to arrive frustrated patients, whilst efficient services, staffed by nurses with the specialist skills needed to manage IV treatment had the opposite effect. Many patients felt a local, GP or community health centre based service would resolve many of the practical difficulties that made OPAT inconvenient. Patients could find OPAT anxiety provoking but this could be ameliorated by staff taking the time to reassure patients, and provide tailored information. (lines 48-62)

Introduction: Para 2 and 3: better placed at start and leading up to aim of this study. Please double check accuracy of referencing: e.g. Ref 11 refers to systematic review but unsure if ref at end list is systematic review. Some references are a little dated.

Response: Thank you for these useful suggestions. We have revised the ordering of the introduction. We have revisited the references and confirm that there was an error in numbering of the references which has been rectified.

We agree that some references are dated, however these were the most recent we could find that made more than passing reference to the topic. This reflects what has been written about the experience of patients on OPAT compared to the utility of drugs and treatment protocols. We have provided a more recent reference relating to patient anxiety relating to OPAT, replacing a 2001 reference (DuBois 2001) with a 2011 paper. (line 110)

Methods: Some more details required here - e.g. who conducted the interviews? Some more details required on ethics - patient confidentiality, safety at home.

Response: We have added more detail about the interviews as requested. This was removed in an earlier draft for brevity.

The interviews were originally conducted as part of a mixed methods study to identify attributes of care which could be used to develop a discrete choice experiment.³ This paper provides a reanalysis of that those data to understand participant experiences and in doing so takes a subtle realist approach which accepts the social world exists independently of our understanding of it, but that it is only accessible via participants experiences and interpretations.¹⁹ (lines 155-160)

Audio recordings were transcribed verbatim, anonymised and managed using NVivo10 software.²⁰ Transcripts were checked to ensure patient confidentiality was maintained and material removed from the transcript which could possibly identify individuals (e.g. name of their doctor, family members (lines 161-164)

Results: More details required describing source of quotes e.g. <65 and >65 may be too broad. This will help contextualise quotes further. Would also help reader determine the broadness and range of participants included.

Response: We have added more information on the ages of participants, e.g. age 40-50.

Results: Timeliness of service: Lines 16-23; need quote to support the report re complaints.

Response: We have added a quote to support the comment about complaints. We have also revised the text slightly to make the nature of these complaints more explicit. See lines 217-225

Convenience and Flexibility

Although some hospital outpatient OPAT services were managed via an appointment system, one NHS Trust ran their OPAT service from the medical admissions ward, leading to significant delays which proved particularly difficult for patients who were trying to balance going to work and treatment. “[I] thought ‘I’ll be back in work by sort of quarter to eleven’, by three o’clock I still hadn’t been seen, [...] I didn’t like that whatsoever” male, 40-50, short term, hospital outpatient

Some quotes very long - may be beneficial to cut down.

Response: We have cut some of the longer quotes as requested. See body text for details (examples include lines 211-214; 236-239; 262-265)

Clarification on which are themes/subthemes particularly in the "delivery of care" section. Also does this relate to functional aspects which are not referred to in results section of text.

Response: We have revised the structure of the results section which we trust makes this clearer. There are two main themes, the first relating to functional and the second to relational aspects of care. The functional aspects of care theme has 4 subthemes, whilst the relational aspects of care theme has been revised in response to the suggestions made by this reviewer. See lines 189-194

Care with bacterial nomenclature = P13. Line 47 - medical equipment breakdown - not relating to OPAT. Unclear whether this influenced patient choice.

Response: We have revised the bacterial nomenclature to *Clostridium difficile* and methicillin-resistant *Staphylococcus aureus* in the body of the text. (lines 297-300)

We have removed the section on the breakdown of equipment as the reviewer is correct that this is not directly related to OPAT, but the storage of drugs.

Relational aspects: There is plenty of overlap between 2 themes. Authors may consider whether more informative to combine the two together. Also aspects such as "dignity" referred to at introduction which do not seem to be followed through themes? Then "These are described by four subthemes: being home, but not well; timeliness of care; location of care; is it safe?"...but there were only three sub-themes in the abstract. On page 14 the theme Relational aspects of care with two sub-themes is presented (in concordance with the abstract)... is that then one of the three main issues around delivery of OPAT...and where is the third?. It seems also that different categories/wording is used. Is Timely management of symptoms (in abstract) for instance, the same as Timeliness of care (in manuscript)? Authors should work through this critically, so it is clear to the reader what the identified themes and sub-themes are and how they inter-relate. I would suggest using headings and subheadings in the result section to make it clear. Also of course consistency between the categories and wording in the abstract and the manuscript.

Response: The reviewer will note that we have completely re-written the relational theme in light of reviewer comments and also revisited the abstract (see earlier) which has also been rewritten. It is unfortunate that some errors crept in during editing and were not picked up and we thank the reviewer for their attention. The concept of dignity is implicit in the way patients are cared for but we have removed specific reference to this at the start of the theme (lines 323- 394.)

Any different between themes from interviews and focus groups?

Response: We found no differences between the focus group and interviews apart from the focus groups offered a more 'descriptive' rather than emotional account of their care. We have added a sentence into the results about this. (lines 181-185)

The focus group participants came from one hospital and all received a nurse at home model so although the issues they identified around nurse at home care reflect the experiences of patients at other centres they did not contribute to our understanding of the other models of care. As a result the findings of the focus group and interviews were analysed with the interview data.

Discussion: Needs expansion. Most is a review summarising key findings. Only ref 26 referred to in appropriate depth. Avoid use of quantitative descriptors when analysing qualitative data - e.g. "over half of patients"; "1/3 of patients"

Response: Very little has been written about this topic in the past beyond the survey data described in our introduction. We have taken the opportunity to expand on these findings and link them to our own findings (lines 404- 416) However, we are also conscious of word length for the paper which has tempered our discussion. We have removed reference to quantitative descriptors of our findings as we recognise these are contentious.

Satisfaction with OPAT services was high, a finding which is well reported in the literature.¹⁰ However, there was also evidence that services were not always well aligned to the personal and material resources of the patient. The contextual factors that affected how well patients cope included: what support families had at home, personal circumstances (e.g. self employed), material resources, such as car access for daily attendance at hospital and the provision of information tailored to their situation. Other studies have found families may not have the personal resources to care for a family member at home^{12 14}, and our findings support this conclusion.

It is recognised that patients often find being cared for out of hospital worrying, and providing access to advice can boost confidence^{12 17}. However, the information needs of patients are often not met.¹⁸ The present study supports these findings, but also suggests that even when patient outcomes are good, as was the case in this study, interpersonal relationships are important.

Strengths: claims that contrasts views of different experiences - this may be clearer in results section. Also needs more discussion on how study was used or may be used to inform practice and policy development

Response: Throughout the results section we have provided examples of how differently patients can experience the same model of care and how personal circumstances influence how care is experienced. A few examples are provided below.

... participants believed that recovering at home would be better than being in hospital, and welcomed the opportunity to try OPAT. However, few realised how difficult it would be to look after themselves at home and some felt staff should have been more alert to their personal situation and circumstances. (lines 198-202)

For others, OPAT was an opportunity to avoid hospital admission, and although some people continued to work, for others, the infection limited their activities. (lines 193-194)

"[I] thought 'I'll be back in work by sort of quarter to eleven', by three o'clock I still hadn't been seen, [...] I didn't like that whatsoever" male, 40-50, short term, hospital outpatient (lines 208-10) which contrasts with the experience of others who found the service convenient.

For patients who self-administered, multiple treatments each day can leave them with little time to fit anything else into their day. Although some coped by taking their IV kit with them and infusing 'on the go', others found the perceived benefit of being at home was eroded, as planning the next treatment was always at the back of their mind. (lines 241-244)

"There's no point really going out much or doing much cos you haven't got much time when you aren't having to think about getting everything sorted" male, age 30-40, long term self-administer

We have also described in more detail how the findings of the qualitative study were used and how these findings may be used by OPAT teams. (lines 392-401)

The findings of this study were used to develop a discrete choice experiment (DCE) to seek to understand patient preferences for OPAT services. The DCE was distributed to 202 people who had previous experience of OPAT and found that looking at the whole sample, patients were more likely to choose a nurse at home model over a hospital or self-administration model; there was a preference for timed appointments, and for treatment delivered by a specialist, rather than generalist nurse, and communication with someone they know. However, there was significant heterogeneity across patient types, albeit with an overall preference for the nurse at home model. ³ These findings align with our qualitative findings, and argue for flexible service as a one-size does not fit all.

REVIEWER #2

OPAT services have the potential to reduce health care cost and improve patients' well-being during treatment, but little is indeed known about the patients' perspective; which model of service they prefer, what are potential challenges and possible improvements. The work presented in this manuscript is indeed important and relevant. I find the manuscript to be very well written, concise and clear. I have only one major and a few minor concerns/suggestion:

Major: To me the structure in the presentation of the findings is very confusing. What are major themes and what are sub-themes?

See response to reviewer 1 – we have revised the results section to make this clearer.

In the abstract it seems functional and relational aspects of care are the two major themes with three and two sub-themes respectively? In the result section however I cannot find this pattern. It says (under a heading called Delivery of care...is this I wonder Functional aspects of care?): "Three main issues around delivery of OPAT were identified...." which are?? Are main issues the same as major

Response: We apologise for the confusion that that original version of the results generated. As already described, we have revisited the manuscript to ensure greater clarity in our terminology and to demonstrate the themes and subthemes more clearly.

Minor:

P. 5 line 12 on the association. Patient experience can be good or bad, is it not rather patient preference?

Response: We have revised this sentence for clarity. The reviewer is correct in suggesting that patient experience can be good or bad, but we would argue that it is broader than patient preference. In most cases the patients in our study did not get a choice of OPAT service, rather the option was OPAT or hospital in-patient. We have used the NICE and the Institute of Medicine definition of patient experience which includes dimensions such as empathy, respect, involvement in decision making, attention to physical needs and the timeliness of care. We have revised the text to clarify this (see lines 95-96)

Across a range of clinical conditions, studies have found positive associations between patient experience (defined by NICE and the Institute of Medicine as the relational and functional aspects of care) 5 6 and a range of quality indicators, including patient safety and clinical effectiveness 7 8

The references – I believe - are 'misplaced'. It looks to me as though a reference was omitted and then the rank order wasn't changed. Authors should review references thoroughly and make sure they are correct, complete and updated.

Response: We thank the reviewer for noticing this and have revised these. See response to reviewer #1

I would also like a few explanatory sentences about the methodological framework chosen and why, before the description of the actual thematic analysis performed

Response: Such a discussion existed in an earlier draft but removed for brevity. We have reinstated this as requested (see lines 155-160)

The interviews were originally conducted as part of a mixed methods study to identify attributes of care which could be used to develop a discrete choice experiment.³ This paper provides a reanalysis of that those data to understand participant experiences and in doing so takes a subtle realist approach which accepts the social world exists independently of our understanding of it, but that it is only accessible via participants experiences and interpretations.¹⁹

VERSION 2 – REVIEW

REVIEWER	Trine Bernholdt Rasmussen Herlev and Gentofte Hospital, Denmark
REVIEW RETURNED	22-Nov-2017
GENERAL COMMENTS	Revisions have indeed improved the manuscript. The study and its findings are interesting and useful, and I will use it in my work with establishing outpatient treatment of patients with infective endocarditis.